# Functional Traits of Herbaceous Plants with Ecological Restoration Potential Under Drought Conditions

**DOI:** 10.3390/plants14233552

**Published:** 2025-11-21

**Authors:** Tong Zou, Yujie Li, Yanling Wu, Qingwen Yang, Shuangcheng Wang, Zhenfu Huang, Qiang Li, Xiaohui Zhou, Tianliang Zheng, Xiangjun Pei, Jingji Li

**Affiliations:** 1State Key Laboratory of Geohazard Prevention and Geoenvironment Protecttion, Chengdu University of Technology, Chengdu 610059, China; lucky00atong@163.com (T.Z.); yangqingwen20@cdut.edu.cn (Q.Y.); 2College of Ecology and Environment, Chengdu University of Technology, Chengdu 610059, China; 15308334637@163.com (Y.L.); wuyanling0312@163.com (Y.W.); xiaohuizhou@cdut.edu.cn (X.Z.); zhengtianliang@cdut.edu.cn (T.Z.); 3Key Laboratory of Synergetic Control and Joint Remediation for Soil & Water Pollution, Ministry of Ecology and Environment, Chengdu University of Technology, Chengdu 610059, China; 4Hydrogeological Environmental Survey Center of the Xinjiang Uygur Autonomous Region Geological Bureau, Urumqi 830011, China; 13399948501@163.com (S.W.); 13629959318@163.com (Z.H.); lq7842@163.com (Q.L.)

**Keywords:** drought stress, functional traits, leaves, roots, SEM, drought resistance evaluation

## Abstract

Herbaceous plant species form the plant synusia of desert ecosystems and play crucial roles in wind-breaking, sand-fixing, and the maintenance of oasis ecosystem stability. This study focused on the leguminous species *Medicago sativa* and *Astragalus laxmannii* and the gramineous species *Elymus dahuricus*, *Agropyron desertorum*, *Agropyron mongolicum* and *Agropyron cristatum*. These are the plants commonly used for ecological restoration in the desertification regions of northwestern China. We conducted a pot experiment with six soil moisture gradients (15% [ck], 12%, 10%, 8%, 6%, and 4%) to simulate drought conditions. We studied how varying degrees of drought stress influenced growth, functional traits and drought resistance, aiming to elucidate the mechanisms by which plants respond to drought. The study indicated that: (1) Drought stress directly and indirectly reduced both the seedling emergence rate and plant height. As soil water content decreased, the seedling emergence rate and plant height declined across all species. (2) The leaves and roots of the plants adopted different strategies to cope with drought stress. As soil water content decreased, leaf water content, leaf succulence, and leaf tissue density increased, while specific leaf area and leaf biomass decreased. Additionally, the root-shoot ratio increased, whereas root biomass, total root length, root surface area, and root volume all decreased. (3) Root system adaptation is the key factor influencing the drought resistance of plants. Drought resistance varied among the six species. *Medicago sativa*, *Elymus dahuricus*, and *Agropyron cristatum* exhibited stronger water retention and more stable growth under drought stress, making them better suited for ecological restoration in arid regions. Our study elucidates differentiation strategies for coping with drought stress and offers essential parameters and theoretical support for species selection and rational assemblage in the ecological restoration of arid regions.

## 1. Introduction

Desertification refers to the phenomenon of land degradation caused by climate change, human activities and other factors in arid, semi-arid and sub-humid areas [1,2]. It remains one of the most severe global ecological problems, jeopardizing both ecological integrity and socio-economic stability in vulnerable areas [3]. China is one of the countries with the most serious desertification in the world [4]. Xinjiang is the province with the largest desertification areas in China. Its desertification land area accounts for as much as 47.7%, with vegetation cover ranging from only 5% to 15% [5,6]. Currently, desertification control primarily relies on engineering measures and plant anti-desertification. Engineering approaches include Herbage-Squared-Sand-Obstacle, *Salix psammophila* sand barriers, and environmentally friendly sand barriers made from bio-materials [7,8]. However, these methods face some challenges such as high costs, susceptibility to burial by sand, and short lifespans [9]. In contrast, plant anti-desertification has become an effective means of preventing desertification due to its long-term ecological benefits [10]. Plants can effectively reduce land degradation risks by covering the soil, trapping sediment particles and reducing wind speed [11]. Additionally, by enhancing the cycling efficiency of elements such as carbon, nitrogen, and phosphorus, plants improve the texture and structure of sandy soils, further enhancing ecosystem productivity and service functions [12].

Ecosystem degradation induced by drought is one of the most prominent features of land desertification [13]. Drought limits water absorption and utilization in plants, resulting in decreased leaf water potential and turgor pressure, which in turn inhibits cell elongation and division and ultimately slows plant growth [14]. Simultaneously, to minimize water loss, plants close their stomata, which in turn restricts carbon dioxide uptake and impairs photosynthesis, leading to reduced productivity [15]. Furthermore, drought stress can induce oxidative damage through the accumulation of reactive oxygen species, which disrupts cellular membranes and metabolic processes [16]. Numerous studies have demonstrated that plants adapt to drought stress through coordinated changes in their morphological, physiological, and biochemical functional traits [17]. Drought stress limits plant height growth, impedes leaf development, and alters root architecture. However, plants can adjust their phenotype and biomass allocation strategies to adapt to drought environments [18]. For instance, the reduction in leaf thickness and specific leaf area increases leaf tissue density, which improves water retention and photosynthetic performance per unit area. Additionally, the reduction in chlorophyll content (per gram of fresh weight) facilitates the reallocation of resources for synthesizing protective osmolytes, thereby enhancing drought stress tolerance [19,20]. Plants also respond to drought stress by optimizing root architecture. This involves reducing total root length and root biomass to limit carbon expenditure and optimize resource allocation. Concurrently, plants increase the root-shoot ratio, allocating more assimilates to the belowground parts, thereby improving water uptake capacity and enhancing overall drought tolerance [21,22]. In desertification control, the selection of drought-adapted native plant species for restoration is crucial [23]. *M. sativa* (*Medicago sativa*) is a high-quality leguminous forage crop widely used for soil improvement and pasture production [24]. *A. laxmannii* (*Astragalus laxmannii*), a typical pioneer plant for windbreaks and sand fixation, plays a key role in stabilizing shifting sand dunes and facilitating the early stages of vegetation establishment [25]. *E. dahuricus* (*Elymus dahuricus*), an important grass species in the northwestern desert-steppe region, has an extensive underground root network, making it valuable for soil and water conservation projects [26]. Additionally, *A. desertorum* (*Agropyron desertorum*), *A. mongolicum* (*Agropyron mongolicum*), and *A. cristatum* (*Agropyron cristatum*) are excellent forage grasses widely distributed in northwestern China; their dense root systems effectively stabilize surface soil [27,28,29].

Previous studies have primarily focused on the physiological indicators of drought resistance or survival thresholds of individual species, with insufficient attention given to the differential adaptation strategies of multiple species based on functional traits. Most studies have emphasized the response of the aboveground parts of plants, while lacking mechanistic explanations of the synergistic changes in both aboveground and underground traits under systematic moisture gradients [30]. To address this gap, this study selected sandy soil from the Hotan Prefecture of Xinjiang as the test substrate and six dominant indigenous herbaceous plant species. Through a pot experiment simulating different moisture gradient conditions, this research focused on investigating drought adaptation strategies and the differences in drought resistance among various species. The study aimed to answer two key scientific questions: (1) How do different plant functional traits change in response to drought stress? (2) What are the morpho-structural strategies and the associated drought resistance mechanism of indigenous herbaceous plants under drought stress? This research provides theoretical support for vegetation restoration and reconstruction in desertification regions, as well as for water management following desert afforestation, and for maintaining ecosystem stability.

## 2. Results

### 2.1. Effects of Drought Stress on Herbaceous Plant Growth

#### 2.1.1. Effects of Drought Stress on Seedling Emergence Rate and Plant Height in Leguminous Species

As soil water content decreased, both leguminous species exhibited declining trends in seedling emergence rate and plant height (Figure 1). For *M. sativa*, at soil water content ≤ 10%, the seedling emergence rate significantly declined (*p* < 0.05). At soil water content ≤ 8%, plant height significantly declined (*p* < 0.05). For *A. laxmannii*, at soil water content ≤ 8%, both seedling emergence rate and plant height showed significant reductions (*p* < 0.05). The two species differed in their sensitivity to drought under the same soil water content conditions. Except for under the 6% and 8% soil water content conditions, *M. sativa* showed a significantly higher seedling emergence rate and plant height than *A. laxmannii* under all other soil water content conditions (*p* < 0.05).

#### 2.1.2. Effects of Drought Stress on Seedling Emergence Rate and Plant Height in Gramineous Species

As the soil water content decreased, the seedling emergence rate and plant height of the four gramineous species exhibited a downward trend, although the degree of decline varied among the species (Figure 2). For *E. dahuricus* and *A. mongolicum*, at soil water content ≤ 10%, both seedling emergence rate and plant height significantly decreased (*p* < 0.05). For *A. cristatum* and *A. desertorum*, at soil water content ≤ 8%, the seedling emergence rate and plant height significantly decreased (*p* < 0.05). Under the CK, 10%, 6%, and 4% soil water content conditions, *E. dahuricus* showed significantly higher plant height than the other three gramineous species (*p* < 0.05). In addition, except under the 6% soil water content condition, *A. cristatum* exhibited significantly higher seedling emergence rates than the other three gramineous species under all other soil water content conditions (*p* < 0.05). These results suggested that *E. dahuricus* and *A. cristatum* exhibited better drought tolerance compared to *A. mongolicum* and *A. desertorum*.

### 2.2. Effects of Drought Stress on Functional Traits of Herbaceous Plants

#### 2.2.1. Effects of Drought Stress on Leaf Functional Traits of Herbaceous Plants

Soil water content significantly influenced the leaf functional traits of the six herbaceous plants. As drought stress intensified, leaf biomass and specific leaf area of all the tested plants showed a downward trend, while leaf water content, degree of succulence, and leaf tissue density increased (Figure 3). These responses varied among species.

For the leguminous plants, at soil water content ≤ 6%, the leaf biomass and specific leaf area of *M. sativa* and *A. laxmannii* decreased significantly (Figure 3a,d: *p* < 0.05). Under the CK, 12%, and 4% soil water content conditions, *M. sativa* exhibited significantly higher leaf biomass than *A. laxmannii* (Figure 3a: *p* < 0.05). At soil water content ≤ 8%, both species showed a significant increase in leaf water content (Figure 3b: *p* < 0.05). At soil water content ≤ 8%, the degree of leaf succulence also significantly increased in both species (Figure 3c: *p* < 0.05). At 6% and 4% soil water content, *M. sativa* had significantly higher leaf tissue density than *A. laxmannii* (Figure 3e: *p* < 0.05).

For the gramineous plants, at soil water content ≤ 6%, the leaf biomass of *E. dahuricus*, *A. desertorum*, *A. mongolicum*, and *A. cristatum* significantly decreased (Figure 3a: *p* < 0.05). At soil water content ≤ 6%, the leaf water content and degree of succulence of *E. dahuricus* and *A. desertorum* significantly increased (Figure 3b,c: *p* < 0.05), whereas *A. mongolicum* and *A. cristatum* showed significant increases in degree of leaf succulence at soil water content ≤ 10%. Under the CK, 12%, 10%, and 8% soil water content gradients, the leaf tissue density of *A. desertorum*, *A. mongolicum*, and *A. cristatum* was significantly higher than that of *E. dahuricus* (Figure 3e: *p* < 0.05).

#### 2.2.2. Effects of Drought Stress on Root Functional Traits of Herbaceous Plants

Soil water content also had a significant effect on root functional traits. As drought stress intensified, the root biomass, total root length, root surface area, and root volume of all tested species declined, whereas the root-shoot ratio increased (Figure 4). Responses differed among species.

For leguminous plants, the root biomass of *M. sativa* significantly decreased when soil water content was ≤6%, while the root biomass of *A. Laxmannii* significantly decreased when soil water content was ≤10% (Figure 4a: *p* < 0.05). The root surface area and root volume of *M. sativa* and *A. Laxmannii* significantly decreased when soil water content was ≤10% (Figure 4d,e: *p* < 0.05).

For the gramineous plants, at soil water content ≤ 6%, root biomass of *E. dahuricus* and *A. cristatum* declined significantly (*p* < 0.05). Under the CK and 12%, 10%, and 8% soil water content conditions, the root biomass of *E. dahuricus* and *A. cristatum* was significantly higher than that of *A. desertorum* and *A. mongolicum* (Figure 4a: *p* < 0.05). In all four gramineous plants, total root length and root surface area declined significantly at soil water content ≤ 8% (Figure 4c,d: *p* < 0.05). Under the CK and 12%, 10%, and 8% soil water content conditions, *E. dahuricus*, *A. cristatum*, and *A. desertorum* exhibited a significantly higher total root length than *A. mongolicum* (Figure 4c: *p* < 0.05). For root volume, *E. dahuricus* and *A. mongolicum* showed significant declines at soil water content ≤ 6%. Under the 12%, 10%, 8%, and 6% soil water content conditions, the root volume of *E. dahuricus* and *A. cristatum* was significantly higher than that of *A. desertorum* and *A. mongolicum* (Figure 4e: *p* < 0.05).

### 2.3. Relative Effects of Drought Stress on Growth Performance and Functional Traits of Herbaceous Plants

To further quantify the relative effects of drought stress on the growth performance and functional traits of herbaceous plants, SEM (Structural Equation Modeling) was developed to elucidate the pathways and interrelationships among these variables (Figure 5a). The results demonstrated significant negative effects of drought stress on both leaf and root functional traits in legumes, with path coefficients of −0.819 (*p* < 0.001) and −0.940 (*p* < 0.001), respectively. Additionally, drought stress also had significant negative effects on both plant height and seedling emergence rate of legumes, with path coefficients of −0.669 (*p* < 0.01) and −0.803 (*p* < 0.001), respectively. For the gramineous plants, similar to the legumes, drought stress had significant negative effects on both leaf and root functional traits, with path coefficients of −0.912 (*p* < 0.001) and −0.948 (*p* < 0.001), respectively. Additionally, drought stress exerted a significant negative effect on the seedling emergence rate of gramineous plants, with a path coefficient of −0.696 (*p* < 0.001). Overall, drought stress exerted negative effects on both growth performance and functional traits. Additionally, drought stress indirectly affected plant height through its influence on the functional traits of both plant leaves and roots. However, the negative effects of drought stress on plant height and seedling emergence rate were weaker in the gramineous plants compared to the leguminous plants (Figure 5b).

### 2.4. Comprehensive Evaluation of Drought Resistance of Herbaceous Plants

Based on PCA, we screened 12 indicators and determined the number of principal components according to the eigenvalue criterion (λ ≥ 1). In the factor loading analysis, variables with higher loadings contributed more strongly to a given principal component. Table 1 shows the following: Mild drought: the first three principal components explained 92.558% of the total variance. PC1 (47.825%) was dominated primarily by leaf biomass and root volume; PC2 (29.026%) by root surface area; PC3 (15.707%) by leaf water content.

Moderate drought: the first four principal components explained 98.423% of the total variance. PC1 (48.123%) was dominated primarily by root volume and root biomass; PC2 (29.703%) by total root length and plant height; PC3 (8.893%) by specific leaf area and leaf water content; and PC4 (1.577%) by the root-shoot ratio.

Severe drought: the first four principal components explained 95.822% of the total variance. PC1 (44.599%) was dominated primarily by root biomass and root volume; PC2 (30.859%) by total root length and plant height; PC3 (11.330%) by the root -shoot ratio; and PC4 (9.034%) by specific leaf area.

According to the results of comprehensive evaluation value (D value) (Table 2), drought resistance is ranked as follows: Mild drought: *E. dahuricus* > *M. sativa* > *A. cristatum* > *A. laxmannii* > *A. desertorum* > *A. mongolicum*; Moderate drought: *A. cristatum* > *E. dahuricus* > *M. sativa* > *A. desertorum* > *A. laxmannii* > *A. mongolicum*; Severe drought: *A. cristatum* > *E. dahuricus* > *M. sativa* > *A. mongolicum* > *A. desertorum* > *A. laxmannii*.

## 3. Discussion

### 3.1. Drought Stress Suppresses Seedling Emergence Rate and Growth by Impacting Physiological Processes and Functional Trait

Seed germination marks the beginning of plant growth and development, a critical phase in the plant life cycle [31,32]. Our study found that, with increasing drought stress, the seedling emergence rate and plant height of both leguminous and gramineous plants were suppressed. Drought stress likely damages the internal membrane systems of seeds, leading to insufficient water absorption, which hinders cell expansion, thereby delaying or inhibiting the germination process [33,34,35]. Water deficit limits photosynthesis and decreases carbohydrates, which may cause the plant height to be suppressed, thereby affecting seedling growth [36]. Furthermore, under drought conditions, the water absorption capacity of the root system declines. The transportation of water through the xylem and phloem is limited, which in turn blocks the cell elongation and division processes, thereby inhibiting normal plant development in height [37]. It is also possible that in water-limited environments, plants may adjust their biomass allocation. The plants enhance root length to obtain more water while reducing aboveground growth, leading to a decrease in plant height [38,39]. SEM analysis further revealed that drought stress negatively affected both growth performance and functional traits, indirectly influencing plant height through leaf and root traits. The suppression of plant height and seedling emergence rate by drought stress was stronger in leguminous plants than in gramineous plants. Such differences likely result from distinct hormonal regulation between legumes and gramineous species [40]. In addition, Legumes typically have relatively shallow root systems and lower root-shoot ratios, resulting in lower water-use efficiency and making them more sensitive to soil water variation. In contrast, gramineous species often possess rapidly expanding fibrous root systems and stronger canopy recovery abilities, allowing them to maintain better water balance and growth performance under drought [41,42].

### 3.2. Root Trait Variation Is a Key Driver of Drought Resistance

Drought resistance in plants is a comprehensive expression of the interaction between genes and the environment. A single indicator cannot fully reflect the strength of drought resistance [43]. Through a comprehensive evaluation of 12 morphological and physiological indicators, this study found significant interspecific differences in drought resistance among the six herbaceous plants. The ranking of the drought resistance was influenced by the intensity of drought. This result affirms the necessity and reliability of multi-indicator comprehensive evaluations in the study of plant stress resistance [44].

PCA revealed that root functional traits were the primary factors driving changes in the principal components, and suggested that root characteristics were key factors influencing drought resistance. Deep and dense root systems were critical pathways for plants to withstand drought stress [45,46]. There are interspecific differences in drought resistance among the herbaceous plants [47]. Our study found that *A. cristatum* and *E. dahuricus* exhibited strong drought resistance under multiple drought stress gradients. Bayat et al. and Jin et al. [48,49] found that the root system of *E. dahuricus* and *A. cristatum* exhibited strong plasticity, maintaining high root-shoot ratios and dense root structures under drought stress. This strategy effectively enhances water acquisition, which contributes to strong drought adaptability. In contrast, *A. desertorum* and *A. mongolicum* exhibited relatively weaker drought resistance, possibly due to limited inherent root morphological plasticity [50,51]. Comparatively, leguminous plants showed intermediate drought resistance, with a decrease in resistance under severe drought conditions. Studies have shown that legumes often allocate a significant portion of their photosynthetic products to symbiotic nitrogen fixation in legume root nodules, which might limit resource allocation for deep root development under drought stress [52]. Additionally, legumes typically have taproots, and their water absorption efficiency from surface soils may be lower than that of gramineous plants which have developed fibrous root systems. Thus, legumes are more susceptible to dramatical reductions in soil water content [53]. The resource allocation trade-off resulting from different biological strategies is a significant contributor to the differences in drought resistance among legumes [54].

### 3.3. Plants Adopt Resource-Conservative and Tolerance Strategies to Cope with Drought Stress

Changes in leaf functional traits can reflect the functional characteristics of the ecosystem in which the plant is located and its ability to adapt to the ecosystem [55]. SEM analysis revealed that drought stress had a significant negative impact on both leaf and root functional traits. Specific leaf area is a crucial indicator of plant functional traits, reflecting a plant’s resource utilization capacity in various habitats [56]. Leaf biomass, the product of photosynthesis, reflects the plant’s external nutrient use efficiency, functioning as a key indicator of adaptation to water variations [57]. We found that, as drought stress increased, the leaf biomass and specific leaf area of both leguminous and gramineous species decreased gradually. The decrease in specific leaf area may be attributed to the closure of stomata under water deficit to decrease transpiration, which suppresses CO_2_ absorption. Thus, photosynthetic capacity is weakened, which reduces carbon assimilation capacity [58]. The reduction in carbon supply not only limits the accumulation of leaf dry matter but also inhibits leaf growth and development [59]. Leaf tissue density is also a crucial indicator for studying the stress resistance of desert plants. Plants with higher leaf tissue density tend to grow more slowly, because they store more carbon for building defense structures, enhancing their ability to conserve water and nutrients, which helps them adapt to arid habitats [60]. Our study found that as soil water content decreased, both leguminous and gramineous species showed a gradual increase in leaf water content, degree of succulence, and leaf tissue density. This may be because drought stress induces morphological adaptations such as increased leaf thickness and succulence, often accompanied by larger or more numerous mesophyll cells and higher palisade tissue density. These structural adjustments enhance the leaf’s water-holding capacity, allowing plants to retain more water and buffer transpirational water loss under drought conditions [61]. Wang et al. [62] also found that degree succulence and water content of leaf increased in response to drought stress. This suggests that plants adapt to drought conditions by reducing specific leaf area, increasing tissue density, and enhancing degree of succulence. This process reduces resource consumption, improves water retention and stress tolerance to improve efficiency in water and nutrient use, forming a resource-conservative strategy [63] (Figure 6).

Roots are an important interface between plants and the soil, transporting water and nutrients to all parts of the plant. Changes in root traits reflect the plant’s adaptation to its environment [64,65]. Numerous studies have demonstrated that when plants experienced drought stress, there was a shift in resource allocation. A greater proportion of assimilates is transported to the roots, resulting in an increase in the root-shoot ratio [66]. In our study, as soil water content decreased, both leguminous and gramineous species showed a gradual increase in the root-shoot ratio. Medeiros et al. [67] observed a consistent result, with an increase in the root-to-shoot ratio under intensified drought stress. This suggests that under drought stress, plants adopt a tolerance strategy. They allocate more biomass to the roots to more effectively utilize limited water resources and withstand severe drought stress [68] (Figure 6). Furthermore, roots are an important source of cytokinins, which coordinate the plant’s overall adaptive response to drought. Therefore, maintaining a high root-shoot ratio and developing a dense root structure improve stress resistance by securing water uptake and maintaining vital hormonal signaling [40]. In our study, as soil water content decreased, both leguminous and gramineous species exhibited a gradual decline in total root length, root surface area, root volume, and root biomass. This result is consistent with the findings of Zhao et al. and Yao et al. [69,70]. This inhibition of root growth may occur because lower soil moisture exceeds the adaptive range of roots [71]. In addition, the leaf’s resource-conservative strategy not only enhances leaf efficiency but also systemically constrains root growth by limiting the supply of photosynthates and auxin. The resulting reduction in resource allocation contributes to the observed declines in root biomass and length, thereby coordinating a whole-plant drought response [61].

## 4. Materials and Methods

### 4.1. Test Site and Materials and Methods

The experiment was conducted at the Teaching Experimental Base of the College of Ecological and Environmental, Chengdu University of Technology, with the temperature of 25 ± 1 °C and the humidity of 65 ± 1%. The trial was carried out from March to May 2024. The test soil was collected from the desert-oasis transition zone in Moyu County, Hotan Prefecture of Xinjiang (79°45′48″–79°46′12″ E, 37°29′48″–37°30′12″ N). The soil type is sandy soil and exhibits typical characteristics of a poor substrate in arid regions, including low water-holding capacity, alkalinity, and low organic matter content. The detailed basic properties of the soil are listed in Table 3. The test plants included *Medicago sativa*, *Astragalus laxmannii*, *Elymus dahuricus*, *Agropyron desertorum*, *Agropyron mongolicum*, and *Agropyron cristatum*. The seeds were purchased from Ningxia Shanggu Agricultural and Animal Husbandry Development Co., Ltd., Ningxia, China.

### 4.2. Experimental Design

This study involved different degrees of drought stress and plant species. The stress levels were designed based on the Chinese agricultural soil drought grade standard (GB/T 32136-2015) [72] and the measured field capacity, by converting the standard’s relative water content ranges into actual soil water content, which guided the establishment of a six-level drought gradient: 4%, 6%, 8%, 10%, 12%, and 15% (CK) (Figure 7). The tested sandy soil was sieved to remove plant roots and debris, then filled into pots with an aperture size of 10.3 cm, a bottom diameter of 8 cm, and a height of 8.5 cm. Each pot contained 450 g of soil, with a layer of gauze placed at the bottom to prevent soil leakage and ensure ventilation. Three days before initiating the water control, the soil was thoroughly watered to reach near-saturation moisture. Afterward, watering was stopped, allowing the soil to naturally dry to the designated gradient. Once the target soil moisture level for each treatment was established and stabilized, seeds were sown, with 50 healthy, plump seeds per pot. After sowing, a 2–3 cm layer of sand was applied to cover the seeds, and the pots were covered with plastic wrap to maintain humidity. The plastic wrap was removed once the seedlings were established. Six soil water gradient treatments were applied to each of the six plant species. Each treatment-species combination was replicated three times, yielding a total of 108 experimental pots. Soil moisture was controlled using the weighing method. Pots were weighed every other day using an electronic balance (precision: 0.1 g). The amount of water loss was determined as the difference between the current pot weight and its target weight. Accordingly, water was added to return the pot to its target weight. This protocol maintained the soil water content within a strict tolerance of ±0.5% of the target value throughout the experiment, ensuring the stability and precision of the moisture gradient [73].

### 4.3. Sample Collection and Analysis

In our study, the primary determination indicators included seedling emergence rate, plant height, leaf fresh weight, leaf surface area, leaf volume, root length, root surface area, root volume, aboveground biomass, and underground biomass. Seedling emergence numbers per pot were recorded each evening for 7 consecutive days after sowing. On the seventh day, seedling emergence rate was calculated. After 60 days of cultivation, harvesting and sampling were performed. For each replicate pot, three individual plants were randomly selected to measure growth and morphological traits. All measurements were conducted at the individual plant level. The height of each plant was measured from the soil surface to the top bud using a measuring tape (accuracy: 1 mm). The entire seedling was then carefully removed from the pot, ensuring the integrity of the root system as much as possible. The seedlings were wrapped in a sealed bag to prevent water loss and promptly taken back to the laboratory. The leaves and roots were separated, and surface impurities were cleaned. The fresh weight of the leaves was immediately measured and recorded. Root and leaf indicators, including leaf surface area, leaf volume, root length, root surface area, and root volume, were analyzed using Epson Expression 11000XL scanner (Epson, Suwa, Nagano, Japan) and the WinRHIZO software (WinRHIZO Arabidopsis 2022a; Regent Instruments, Quebec, Canada) [74,75]. After scanning, the leaves and roots were packed in envelope bags and placed in an oven at 75 °C for 48 h until constant weight was reached. Finally, the dry weights of the different plant parts were measured using an analytical balance. In addition, the field capacity of the soil was determined using the standard cutting ring method. Air-dried substrate was passed through a 1 mm mesh and packed into cutting rings. The rings were saturated by immersion in water for 24 h and then allowed to drain freely for 8 h. Subsequently, the water content of the drained substrate was determined gravimetrically (oven-drying at 105 °C to constant weight). The mean value from replicates was recorded as the field capacity [76]. The calculation methods for each indicator are shown in Table 4 [77,78,79].

### 4.4. Data Processing and Analysis

Data processing and statistical analysis were performed using SPSS 26.0. Principal component analysis (PCA) was employed to reduce the dimensionality of leaf and root traits across different plant families. Structural Equation Modeling (SEM) was conducted using the “lavaan” package in R version 4.4.2 to analyze the relative impact of drought stress on the physiological traits of different plants. One-way ANOVA and *t*-test was applied to evaluate the variance in plant growth, leaf traits, and root traits of the six desertification-prone herbaceous species under different soil moisture conditions (95% confidence interval). The results were expressed as means ± standard error, and multiple comparisons were conducted using LSD. Graphs were generated using GraphPad Prism 9.5.

Based on the combination of PCA and membership function values, drought resistance of the six herbaceous plant species was comprehensively evaluated. The uncertainty of both the weights derived from Principal Component Analysis and the subsequent D-value rankings was quantified using a Bootstrap resampling approach with 1000 iterations. The standard errors and 95% confidence intervals for PCA loadings and D-value rankings are reported in Appendix A Table A1, Table A2, Table A3, Table A4, Table A5 and Table A6. In accordance with the Chinese Agricultural Soil Drought Classification Standard (GB/T 32136-2015) [72], the drought levels in this experiment were defined by converting the thresholds for mild, moderate, and severe drought from relative soil moisture to actual soil water content. The results of the classification are as follows: mild drought (soil water content of 12% and 15%), moderate drought (soil water content of 8% and 10%), and severe drought (soil water content of 4% and 6%). On the basis of PCA, the membership function values μ(x) for all comprehensive indicators were calculated (Formula (1)). The weights of the comprehensive indicators were calculated based on their contribution to the comprehensive indicator value of principal component (Formula (2)). Finally, the comprehensive evaluation value (D value) was calculated (Formula (3)). According to the D values, the drought resistance of the six herbaceous species under different drought degrees was ranked. The larger the D value, the stronger the plant’s drought resistance [80].

The fuzzy membership function:(1)$$\mu \left(X_{i}\right)=\frac{X_{i}-X_{imin}}{X_{imax}-X_{imin}},\;\;i\;=\;1,\;2,\;\ldots ,\;n$$

*X_i_* represents the value of the *i*-th indicator. X*_imin_* represents the minimum value of the *i*-th indicator; X*_imax_* represents the maximum value of the *i*-th indicator.

Weights of the comprehensive index:(2)$$W_{j}=\frac{V_{j}}{\sum_{j=1}^{m}V_{j}}$$

*Vj* is the contribution rate of the *j*-th comprehensive indicator.

Drought resistance comprehensive evaluation value:(3)$$D=\sum_{j=1}^{n}\left[\mu \left(X_{j}\right)W_{j}\right],\;\;j\;=\;1,\;2,\;\ldots ,\;n$$

## 5. Conclusions

This study provides insights into the responses of herbaceous plant functional traits to drought stress and their drought resistance mechanisms. Drought stress was found to directly and indirectly suppress seedling emergence and plant growth. To cope with this stress, plants employ a sophisticated root-leaf coordination strategy. Leaves adopt a resource-conservative strategy, enhancing water-use efficiency and retention capacity through structural regulation. Meanwhile, roots implement a tolerance strategy, improving water acquisition through biomass reallocation and maintaining a relatively higher root–shoot ratio despite the overall biomass reduction. These findings reflect relative rather than absolute trait adjustments, highlighting the plants’ adaptive balance between resource conservation and stress tolerance. Our results confirm that root plasticity serves as a critical factor determining plant drought resistance. Based on a comprehensive evaluation of multiple trait indicators, *E. dahuricus*, *A. cristatum*, and *M. sativa* emerge as highly promising candidates for ecological restoration in arid regions. However, it is important to acknowledge that these findings are derived from controlled pot experiments, where factors such as restricted soil volume and artificial moisture gradients may affect plant responses. We therefore recommend that future field trials be conducted to validate the performance and ecological adaptability of these species under natural conditions before proceeding with large-scale restoration efforts.

## Figures and Tables

**Figure 1 plants-14-03552-f001:**
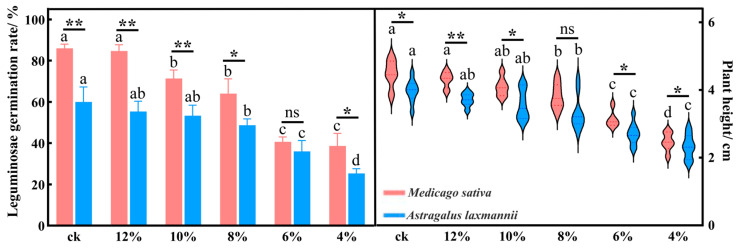
Changes in seedling emergence rate and plant height of leguminous plants under drought stress. Note: Different lowercase letters indicate significant differences within the same species under different soil water content conditions (one-way ANOVA, LSD test, *p* < 0.05). * *p* < 0.05 and ** *p* < 0.01 indicate significant differences between different species under the same soil water content condition, while ns indicates no significant difference (independent samples *t*-test).

**Figure 2 plants-14-03552-f002:**
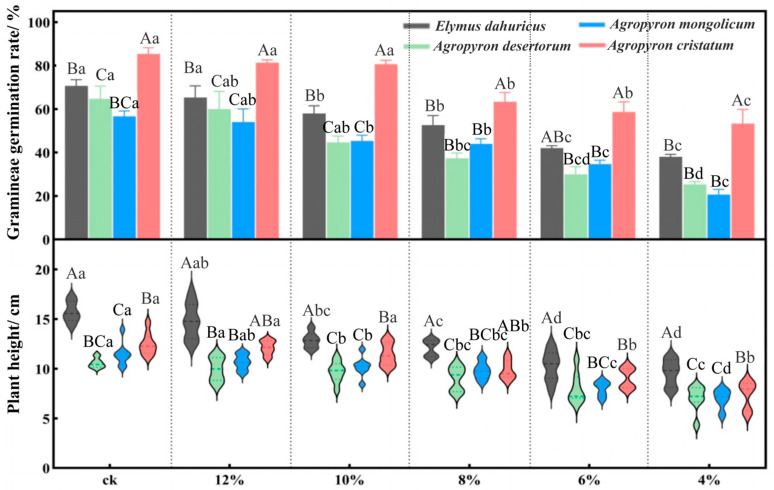
Changes in seedling emergence rate and plant height of leguminous plants under drought stress. Note: Different lowercase letters indicate significant differences within the same species under different soil water content conditions (one-way ANOVA, LSD test, *p* < 0.05). Different uppercase letters indicate significant differences between different species under the same soil water content condition (one-way ANOVA, LSD test, *p* < 0.05).

**Figure 3 plants-14-03552-f003:**
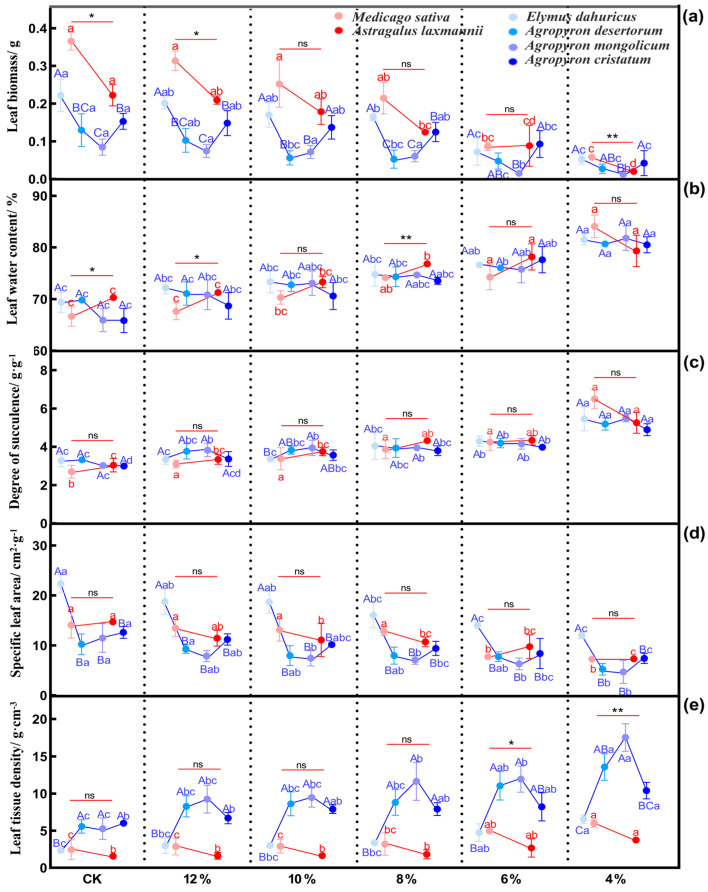
Changes in leaf functional traits of herbaceous plants under drought stress. (**a**–**e**) represent changes in leaf biomass, leaf water content, degree of succulence, specific leaf area, and leaf tissue density, respectively. Note: Different lowercase letters indicate significant differences within the same species under different soil water content conditions (one-way ANOVA, LSD test, *p* < 0.05). Different uppercase letters indicate significant differences in the gramineous plants under the same soil water content condition (*p* < 0.05). * *p* < 0.05 and ** *p* < 0.01 indicate significant differences in the leguminous plants under the same soil water content condition, while ns indicates no significant difference (independent samples *t*-test).

**Figure 4 plants-14-03552-f004:**
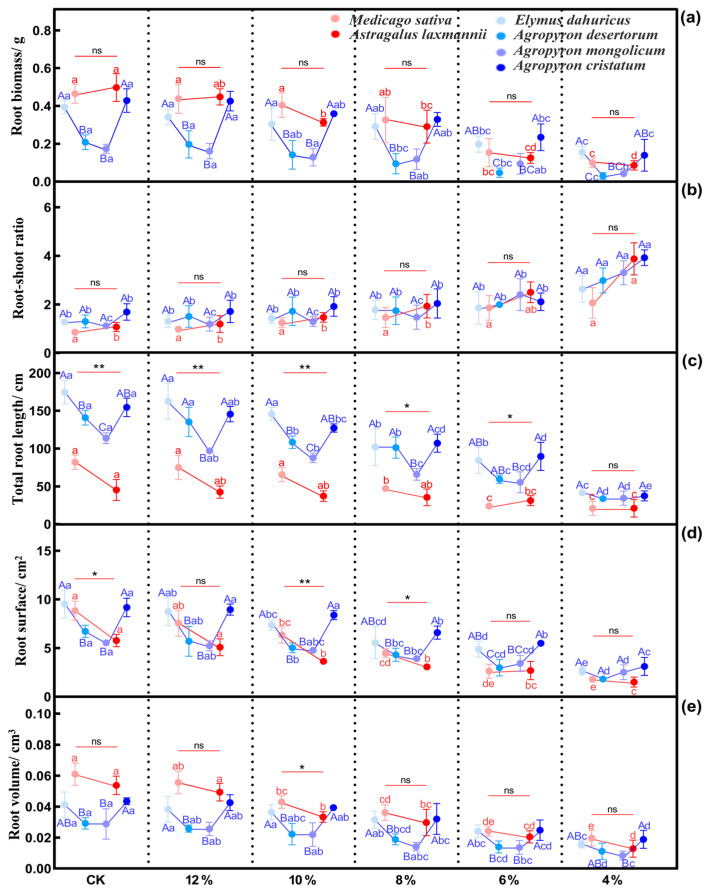
Changes in root functional traits of herbaceous plants under drought stress. (**a**–**e**) represent changes in root biomass, root-shoot ratio, total root length, root surface, and root volume, respectively. Note: Different lowercase letters indicate significant differences within the same plant under different soil water content conditions (one-way ANOVA, LSD test, *p* < 0.05). Different uppercase letters indicate significant differences in the gramineous plants under the same soil water content condition (*p* < 0.05). * *p* < 0.05 and ** *p* < 0.01 indicate significant differences in the leguminous plants under the same soil water content condition, while ns indicates no significant difference (independent samples *t*-test).

**Figure 5 plants-14-03552-f005:**
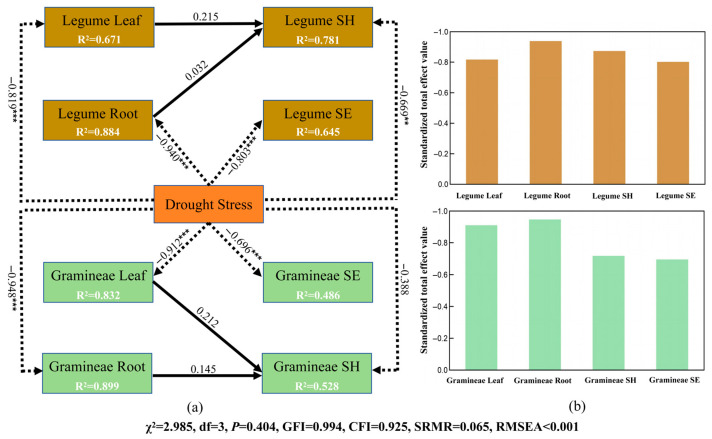
Pathways and standardized total effects of drought stress on the growth performance and functional traits of herbaceous plants. (**a**) SEM. Leaf, Root, SH, and SE represent leaf traits, root traits, plant height, and seedling emergence rate, respectively. The black solid lines and black dashed lines indicate positive and negative effects, respectively. The numbers on the lines represent the path coefficients: ** indicating *p* < 0.01; *** indicating *p* < 0.001. χ^2^ (assessing the discrepancy between the sample and fitted covariance matrices), df (reflecting the model complexity), χ^2^/df indicates the overall goodness of fit of the model. A χ^2^/df value < 3, together with a nonsignificant χ^2^ test (*p* > 0.05), indicates that the model fits the data well. GFI (Goodness of Fit Index) and CFI (Comparative Fit Index) > 0.90, indicate that the model fits well. SRMR (Standardized Root Mean Square Residual) < 0.08, indicates that the model fits well. RMSEA (Root Mean Square Error of Approximation) < 0.05 indicates that the model fits well; (**b**) The standardized total effect of drought stress on the growth performance and functional traits of herbaceous plants.

**Figure 6 plants-14-03552-f006:**
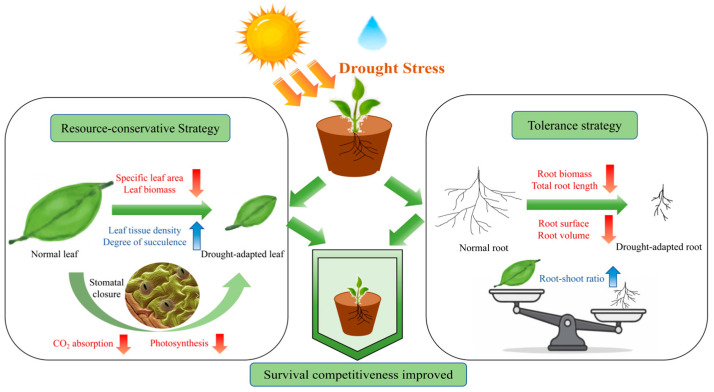
Different response strategies of plant leaves and roots under drought stress.

**Figure 7 plants-14-03552-f007:**
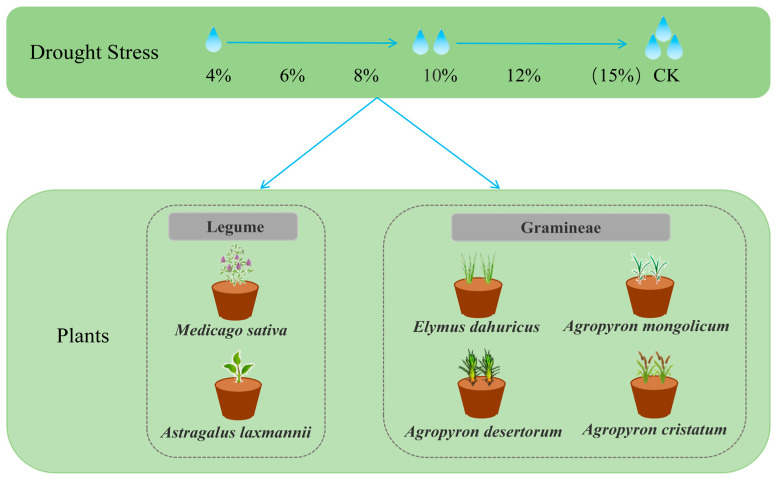
The experimental design included six soil water content gradients to simulate different degrees of drought stress. Each treatment was applied to six plant species, including two leguminous species (*Medicago sativa* and *Astragalus laxmannii*) and four gramineous species (*Elymus dahuricus*, *Agropyron desertorum*, *Agropyron mongolicum*, and *Agropyron cristatum*).

**Table 1 plants-14-03552-t001:** Loadings and explained variance of each indicator in PCA under different drought stress conditions.

Indicator	Mild Drought			Moderate Drought				Severe Drought			
	PC1	PC2	PC3	PC1	PC2	PC3	PC4	PC1	PC2	PC3	PC4
Seedling emergence rate	0.132	0.464	−0.241	0.380	0.069	−0.257	0.129	0.340	−0.071	0.179	0.502
Plant height	−0.290	0.347	0.183	−0.053	0.462	0.343	−0.161	0.064	0.467	0.307	−0.197
Leaf biomass	0.394	0.026	0.071	0.365	−0.236	0.003	−0.168	0.389	−0.135	−0.148	0.052
Leaf water content	−0.098	−0.146	0.630	−0.228	−0.270	0.537	0.111	0.372	−0.041	0.089	0.161
Degree of succulence	−0.391	−0.071	0.219	−0.343	−0.224	0.120	0.292	0.083	−0.425	0.398	−0.141
Specific leaf area	0.181	0.259	0.529	0.293	−0.074	0.539	−0.266	0.297	0.099	−0.053	−0.671
Leaf tissue density	−0.353	0.053	−0.374	−0.250	0.360	−0.332	−0.087	−0.279	0.257	0.386	0.352
Root-shoot ratio	−0.215	0.353	−0.117	0.080	0.219	0.143	0.838	−0.155	0.189	−0.700	0.187
Root biomass	0.372	0.138	0.009	0.392	−0.126	−0.005	0.190	0.403	0.142	−0.147	0.027
Total root length	−0.222	0.425	0.145	0.047	0.488	0.301	−0.037	0.101	0.495	0.123	−0.066
Root surface	0.138	0.493	0.035	0.275	0.395	0.034	0.005	0.214	0.440	0.033	0.142
Root volume	0.413	0.023	−0.059	0.405	−0.103	0.013	0.112	0.418	−0.077	−0.082	0.157
Eigenvalue	5.739	3.483	1.885	5.775	3.564	1.405	1.067	5.352	3.703	1.36	1.084
Contribution rate/%	47.825	29.026	15.707	48.123	29.703	11.705	8.893	44.599	30.859	11.33	9.034
Cumulative contribution rate/%	47.825	76.851	92.558	48.123	77.826	89.531	98.423	44.599	75.458	86.788	95.822

**Table 2 plants-14-03552-t002:** Comprehensive index values, membership function values, and rankings of six herbaceous plants under drought stress.

Drought Degree	Species	Comprehensive Indicator Value				Membership Function Value				D Value	Rank
		Cl_1_	Cl_2_	Cl_3_	Cl_4_	μ_1_	μ_2_	μ_3_	μ_4_		
Mild drought	*Medicago sativa*	3.443	−0.344	−0.869	-	1	0.364	0.181	-	0.661	2
	*Astragalus laxmannii*	1.852	−2.050	0.660	-	0.742	0	0.579	-	0.482	4
	*Elymus dahuricus*	0.105	1.895	2.279	-	0.459	0.841	1	-	0.671	1
	*Agropyron desertorum*	−2.728	−0.727	0.230	-	0	0.282	0.467	-	0.168	5
	*Agropyron mongolicum*	−2.429	−1.415	−0.736	-	0.048	0.135	0.216	-	0.104	6
	*Agropyron cristatum*	−0.242	2.641	−1.564	-	0.403	1	0	-	0.522	3
Moderate drought	*Medicago sativa*	2.712	−1.431	−1.226	−0.973	1	0.291	0	0.004	0.577	3
	*Astragalus laxmannii*	−0.633	−2.961	0.484	1.205	0.416	0	0.514	1	0.355	5
	*Elymus dahuricus*	1.492	0.743	2.097	−0.717	0.787	0.704	1	0.121	0.727	2
	*Agropyron desertorum*	−2.473	1.042	−0.033	0.295	0.095	0.761	0.359	0.584	0.372	4
	*Agropyron mongolicum*	−3.017	0.309	−0.600	−0.981	0	0.622	0.188	0	0.210	6
	*Agropyron cristatum*	1.919	2.298	−0.722	1.172	0.862	1	0.152	0.985	0.830	1
Severe drought	*Medicago sativa*	1.761	−3.044	0.983	0.633	0.893	0	1	0.748	0.604	3
	*Astragalus laxmannii*	−0.737	−1.492	−2.052	−0.603	0.399	0.301	0	0.336	0.314	6
	*Elymus dahuricus*	2.301	1.334	0.601	−1.613	1	0.849	0.874	0	0.842	2
	*Agropyron desertorum*	−2.530	0.246	0.567	−0.133	0.044	0.638	0.863	0.493	0.375	5
	*Agropyron mongolicum*	−2.753	0.843	0.635	0.326	0	0.754	0.886	0.646	0.408	4
	*Agropyron cristatum*	1.958	2.113	−0.734	1.389	0.932	1	0.434	1	0.901	1

**Table 3 plants-14-03552-t003:** Physicochemical properties of the experimental sandy soil. Note: OM: organic matter; TN: total nitrogen; TP: total phosphorus; TK: total kalium: AN: alkali hydrolysis nitrogen: AP: available phosphorus: AK: available kalium.

Indicator	Water Content %	pH	OM g·kg^−1^	TN g·kg^−1^	TP g·kg^−1^	TK g·kg^−1^	AN mg·kg^−1^	AP mg·kg^−1^	AK mg·kg^−1^
Value	0.33	8.2	2.14	1.09	0.31	1.87	12.33	0.52	125.7

**Table 4 plants-14-03552-t004:** Methods for measuring plant growth parameters, leaf and root traits.

	Indicator	Unit	Measurement Method
Growth	seedling emergence rate	%	Emergence Number/Seed Number × 100%
	plant height	cm	Measurement with a ruler (precion:1mm).
Leaf	leaf biomass	g	Dry in a 75 °C oven for 48 h and weigh
	leaf water content	%	(Leaf Fresh Weight-Leaf Dry Weight)/Leaf Fresh Weight × 100%
	degree of succulence	g·g^−1^	Leaf Fresh Weight/Leaf Dry Weight
	specific leaf area	cm^2^·g^−1^	Leaf Surface Area/Leaf Dry Weight
	leaf tissue density	g·cm^−3^	Leaf Biomass/Leaf Volume
Root system	root biomass	g	Dry in a 75 °C oven for 48 h and weigh
	root-shoot ratio	/	Underground Biomass/Aboveground Biomass
	total root length, root surface, root volume	cm, cm^2^, cm^3^	Root Scanner: Epson Expression 11000XL

## Data Availability

The original contributions presented in the study are included in the article; further inquiries can be directed to the corresponding author (Jingji Li, lijingji2014@cdut.edu.cn).

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
