# Peer review of "Functional Traits of Herbaceous Plants with Ecological Restoration Potential Under Drought Conditions"

_plants, 2025, doi:10.3390/plants14233552_

Round 1
Reviewer 1 Report
Comments and Suggestions for Authors
Title: I suggest reformulating the title to better represent the content of the manuscript. For example, it is important to state that the species have ecological restoration potential under desertification conditions. Something like: Drought Resistance of Herbaceous Plants with ecological restoration potential.
Abstract: Please, provide a comprehensive conclusion: which species is more adequate for ecological restoration and why?
Introduction
Please, better describe the species studied, providing a justification for using them.
A more in-depth revision about the effects of drought in plants must be provided: It is known that it reduces growth and productivity, but how does that occur?
How does the adaptive strategies cited in L62-74 can improve drought tolerance? Please explain the mechanisms in details.
Material and methods
L358: What does that percentage mean? How these values were chosen/determined? Please, better explain. Is 15% a suitable control? Isn’t it too low? Please, justify the choice for these drought conditions.
All figures and tables must be self-explanatory. Please, add all the information required to fully understand them.
Results
In the figures, usually the highest value is represented by A, being followed by B and C in the lowest values. Please, rearrange.
Describe the statistical test used in each figure/table
How authors explain the increase in leaf water content under increasing drought?
All results section must be revised, as authors often state significant differences where it does not occur. For instance, in L132-133, when authors say “Except for the 6% soil water content condition, M. sativa showed higher leaf biomass than A. laxmannii”, it is not true, as the figure shows no significance also under 10% and 8%. Same occurs in lines 139-140 and 144-145. Revise the entire section, including the cases I did not mention here. After the case in lines 163-164, I stopped highlighting it in the text, but these misinterpretations are frequent.
Please, in the text, replace all “<” symbols by “≤” to avoid ambiguity. For example, when you say <10%, it is not clear if you refer to 8%, 6%, and 4% only, or if you include 10% in this. This can be avoided by adding the ≤ symbol.
All acronyms must be described the first time they appear in the text (L189) and also in the figure legends.
Discussion and conclusion
Authors must better justify the choice for the drought conditions, and how they determined they were mild, moderate, or severe.
Please, reformulate the conclusion avoiding the topics form. It may be a direct and objective text instead of a summary of your results. I suggest adding this summary to another topic in the discussion section, and creating a real conclusion that answer the original question/hypothesis of the study.

Author Response
Comments 1: Title: I suggest reformulating the title to better represent the content of the manuscript. For example, it is important to state that the species have ecological restoration potential under desertification conditions. Something like: Drought Resistance of Herbaceous Plants with ecological restoration potential.
Response 1: Thank you for your valuable feedback. We agree with this comment and have revised the title to better reflect the focus of the manuscript, including the ecological restoration potential of the herbaceous plants under drought conditions. The updated title is: " The Responses of Functional Traits and Drought Resistance Evaluation in Herbaceous Plants with Ecological Restoration Potential under Drought Stress " This change can be found in the title of the revised manuscript (Page 1, Line 2). We hope that this revision aligns with the intended focus of the study and improves the clarity of the manuscript.
Comments 2: Abstract: Please, provide a comprehensive conclusion: which species is more adequate for ecological restoration and why?
Response 2: Thank you for your insightful suggestion to provide a more comprehensive conclusion. We have revised the abstract to include a clearer conclusion. The revised abstract now reads: Drought resistance varied among the six species. Medicago sativa, Elymus dahuricus, and Agropyron cristatum exhibited stronger water retention and more stable growth under drought stress, making them better suited for ecological restoration in arid regions. We hope this revision provides a more comprehensive summary of our findings. Thank you again for your valuable feedback, which has significantly improved the clarity and completeness of our manuscript. (Page 1, Line 35-39)
Comments 3: Introduction: Please, better describe the species studied, providing a justification for using them.
Response 3: Thank you for this valuable suggestion. We agree that a stronger justification for the selection of our study species will enhance the introduction. We have revised the Introduction to provide a more detailed description of the six native herbaceous species used in this study and to explain their selection, and have added supporting references [23] in this context. We believe these revisions have significantly strengthened the logical flow and scientific rationale of our introduction. (Page 2, Line 83-93)
[23] Kozub, P.C.; Cavagnaro, J.B.; Cavagnaro, P.F. Exploiting genetic and physiological variation of the native forage grass Trichloris crinita for revegetation in arid and semi-arid regions: An integrative review. Grass Forage Sci. 2018, 73, 257–271.
Comments 4: Introduction: A more in-depth revision about the effects of drought in plants must be provided: It is known that it reduces growth and productivity, but how does that occur?
Response 4: Thank you for this valuable comment. We agree that a more detailed explanation of the physiological mechanisms underlying drought's impact on plants was necessary. In the revised Introduction, we provide a more comprehensive and mechanistic description of how drought stress reduces plant growth and productivity, along with supporting references [14-17]. These additions provide a clearer explanation of how drought limits plant growth and productivity. We believe this revision has significantly strengthened the scientific foundation of our introduction. Thank you again for your helpful feedback. (Page 2, Line 64-72 and Page 19, Line 569-576)
[14] Xu, C.; Ke, Y.G.; Zhou, W.; Luo, W.T.; Ma, W.; Song, L.; Smith, M.D.; Hoover, D.L.; Wilcox, K.R.; Fu, W.; et al. Resistance and resilience of a semi-arid grassland to multi-year extreme drought. Ecol. Indic. 2021, 131, 108139.
[15] Cheng, Z.; Qian, Z.H.; Huang, B.C.; Feng, G.L.; Sun, G.Q. Increased drought impacts on vegetation productivity in drylands under climate change. Geophys. Res. Lett. 2025, 52, e2025GL115616.
[16] Sato, H.; Mizoi, J.; Shinozaki, K.; Yamaguchi-Shinozaki, K. Complex plant responses to drought and heat stress under climate change. Plant. J. 2024, 117, 1873–1892.
[17] Bufford, J.L.; Hulme, P.E. Increased adaptive phenotypic plasticity in the introduced range in alien weeds under drought and flooding. Biol. Invasions 2021, 23, 2675–2688.
Comments 5: Introduction: How does the adaptive strategies cited in L62-74 can improve drought tolerance? Please explain the mechanisms in details.
Response 5: Thank you for this insightful comment, which has helped us significantly improve the mechanistic depth of our introduction. As suggested, we have revised the section to provide a detailed explanation of how the cited adaptive strategies enhance drought tolerance. We have expanded on the physiological mechanisms underlying these adaptive strategies. The revised text explicitly links each morphological trait change to its underlying physiological function. We believe these additions provide the required mechanistic detail and create a much stronger foundation for our study. (Page 2, Line 63-83)
Comments 6: Material and methods: L358: What does that percentage mean? How these values were chosen/determined? Please, better explain. Is 15% a suitable control? Isn’t it too low? Please, justify the choice for these drought conditions.
Response 6: Thank you for these critical questions. In our study, the percentages refer to the Actual Soil Water Content (ASWC). The drought gradient (4%, 6%, 8%, 10%, 12%, and 15%) was rigorously designed based on the Chinese agricultural soil drought grade standard (GB/T 32136-2015) and our measured Field Capacity (determined via the cutting ring method).
The standard defines drought by Relative Water Content (RWC = ASWC /FC). We converted its thresholds for severe (RWC 25-35%), moderate (RWC 35-45%), and mild (RWC 45-55%) drought into Actual Soil Water Content ranges of severe (ASWC 5.59%-7.83%), moderate (ASWC 7.83%-10.07%), and mild (RWC 10.07%-12.31%), respectively. Based on these actual moisture content thresholds, we established the soil moisture gradients as 4%, 6%, 8%, 10%, 12%, and 15% (with 15% as the control group, CK).
Furthermore, it is important to note that we included the 4% and 15% ASWC in our drought gradient, which extend beyond the defined thresholds. The 4% ASWC was included below the severe drought threshold to test the absolute survival limits of the plants, while the 15% ASWC was set above the mild drought upper limit to establish an unequivocal and well-watered control baseline. We have revised the 'Materials and Methods' section to clarify the criteria for the drought gradient, and have added a supporting reference [69] for the determination of field capacity. We hope this clarification addresses your concerns and provides a solid scientific rationale for our experimental design. (Page 12, Line 390-394 and Page 14, Line 437-442, 460-466)
[69] Wu,P.; Wang, Y.; Li, Y.; Yu, H.; Shao, J.; Zhao, Z.; Qiao, Y.; Liu, C.; Liu, S.; Gao, C. Optimizing irrigation strategies for sustainable crop productivity and reduced groundwater consumption in a winter wheat-maize rotation system. J. Environ. Manag. 2023, 348, 119469.
Comments 7: Material and methods: All figures and tables must be self-explanatory. Please, add all the information required to fully understand them.
Response 7: Thank you for this insightful suggestion. We have carefully revised figure and table captions in the manuscript to ensure they are self-explanatory. We hope these modifications appropriately address the comment and enhance the clarity of our presented data. (Page 13, Line 414-417 and Page 14, Line 444)
Comments 8: Results: In the figures, usually the highest value is represented by A, being followed by B and C in the lowest values. Please, rearrange.
Response 8: Thank you for your valuable comment. We have rearranged the figures as requested, with the highest values now represented by "A," followed by "B" and "C" in descending order. The revised figures have been updated accordingly. (Page 4, Figure 2, Page 5, Figure 3 and Page 6, Figure 4)
Comments 9: Results: Describe the statistical test used in each figure/table.
Response 9: Thank you for your helpful comment. We have added a description of the statistical tests used in each figure and table. We hope these modifications meet the requirements and enhance the clarity of our results presentation. (Page 3, Line 126-128, Page 4, Line 146-148 Page 5, Line 174-177, Page 7, Line 203-206)
Comments 10: Results: How authors explain the increase in leaf water content under increasing drought?
Response 10: Thank you for this insightful question. In our study, the observed increase in leaf water content under increasing drought is likely related to the plants' morphological adaptations. Drought stress often triggers morphological changes, including an increase in leaf thickness or a shift towards succulence. This is frequently associated with larger or more numerous mesophyll cells and an increase in palisade tissue density. These changes led to structural modifications that enhanced water retention, enabling the plant to store more water and thereby buffer transpirational water loss. As a result, an increase in leaf water content was observed under drought stress. We have revised the Discussion section to include this explanation, and have added a supporting reference [55]. we hope that this addition has enhanced the clarity of our findings. (Page 11, Line 335-341)
[55] Gupta, A.; Rico-Medina, A.; Caño-Delgado, A.I. The Physiology of Plant Responses to Drought. Science 2020, 368, 266–269.
Comments 11: Results: All results section must be revised, as authors often state significant differences where it does not occur. For instance, in L132-133, when authors say “Except for the 6% soil water content condition, M. sativa showed higher leaf biomass than A. laxmannii”, it is not true, as the figure shows no significance also under 10% and 8%. Same occurs in lines 139-140 and 144-145. Revise the entire section, including the cases I did not mention here. After the case in lines 163-164, I stopped highlighting it in the text, but these misinterpretations are frequent.
Response 11: We sincerely thank the reviewer for this critical comment. We carefully rechecked all figures and statistical results in the Results section. We found that in some cases, the differences previously described as significant were not supported by the statistical analysis. We have now corrected these inaccuracies and revised all relevant sentences and other similar cases throughout the section. We believe that this comprehensive revision has eliminated these inaccuracies. We are truly grateful for the reviewer's vigilance, which has been essential in improving the accuracy and integrity of our manuscript. (Page 4, Line135-138 156-158, 161-162, Page 5, Line 168-170, Page 6, Line 189-191,196-199)
Comments 12: Results: Please, in the text, replace all “<” symbols by “≤” to avoid ambiguity. For example, when you say <10%, it is not clear if you refer to 8%, 6%, and 4% only, or if you include 10% in this. This can be avoided by adding the ≤ symbol.
Response 12: Thank you for this helpful suggestion. We agree that using “≤” instead of “<” improves clarity and avoids ambiguity in the description of soil water content levels. Accordingly, we have carefully revised the text and replaced all instances of “<” with “≤” throughout the manuscript to ensure consistency and precision.
Comments 13: Results: All acronyms must be described the first time they appear in the text (L189) and also in the figure legends.
Response 13: Thank you for pointing out this oversight. We have now defined the acronym "SEM" (Structural Equation Modeling) upon its first appearance in the main text and have also ensured that all other acronyms in the manuscript and figure legends are properly defined when first used. (Page 7, Line210)
Comments 14: Discussion and conclusion: Authors must better justify the choice for the drought conditions, and how they determined they were mild, moderate, or severe.
Response 14: Thank you for your comment. As previously explained in our response to Comment 6, the drought conditions were Determined based on the Chinese Agricultural Soil Drought Classification Standard (GB/T 32136-2015) and the measured field capacity. The relative water content (RWC) thresholds for severe, moderate, and mild drought were converted into actual soil water content (ASWC) ranges, and the drought gradients for mild, moderate, and severe drought were accordingly determined. We hope this addresses your concerns, and we have enhanced the Materials and Methods section (4.2 Experimental Design
and 4.4 Data Processing and Analysis) accordingly to further clarify these points. (Page 12, Line 390-394 and Page 14, Line 460-466)
Comments 15: Discussion and conclusion: Please, reformulate the conclusion avoiding the topics form. It may be a direct and objective text instead of a summary of your results. I suggest adding this summary to another topic in the discussion section, and creating a real conclusion that answer the original question/hypothesis of the study.
Response 15: We sincerely thank you for this critical suggestion. We have completely restructured the final section of our manuscript in accordance with your advice. The results has been integrated into the Discussion section to support our arguments. In its place, we have created a new, concise Conclusion that directly addresses the original research questions by synthesizing the broader implications of our findings and providing a definitive answer regarding plant drought resistance mechanisms and species selection for ecological restoration. The revised Conclusion now states:
This study provides insights into the responses of herbaceous plant functional traits to drought stress and their physiological resistance mechanisms. Drought stress was found to directly and indirectly suppress seedling emergence and plant growth. To cope with this stress, plants employ a sophisticated root-leaf coordination strategy. Leaves adopt a resource-conservative strategy, enhancing water-use efficiency and retention capacity through structural regulation. Meanwhile, roots implement a tolerance strategy, improving water acquisition through biomass reallocation and maintaining a relatively higher root–shoot ratio despite the overall biomass reduction. These findings reflect relative rather than absolute trait adjustments, highlighting the plants’ adaptive balance between resource conservation and stress tolerance. Our results confirm that root plasticity serves as a critical factor determining plant drought resistance. Based on a comprehensive evaluation of multiple trait indicators, E. dahuricus, A. cristatum, and M. sativa emerge as highly promising candidates for ecological restoration in arid regions. However, it is important to acknowledge that these findings are derived from controlled pot experiments, where factors such as restricted soil volume and artificial moisture gradients may affect plant responses. We therefore recommend that future field trials be conducted to validate the performance and ecological adaptability of these species under natural conditions before proceeding with large-scale restoration efforts. (Page 15, Line 482-499)

Reviewer 2 Report
Comments and Suggestions for Authors
The current paper studied differential adaptation strategies to drought stress in multiple species based on functional traits. Authors include in analysis Medicago, Agropyron, Elimus and Astragalus and studied it functional traits under different drought conditions in order to use it for soil desertification areas.
Line 47: development of human society- may be too strong conclusion…
Line 68: “also” - to what also??
Line 69: “inhibited chlorophyll synthesis” per leaf? Per FW?
Line 79 “traits respond to drought” – traits itself can not respond.
Line 86: resistance mechanism- which one- physiological? Epigenetic?
Line 96: Seedling germination?? Seedling can not germinate- seeds can as chromatin activation and primary gradients establishment.
Line 104: Changes in germination rate and plant height- at which time point plant height? 3 days? 10 days? What is germination rate – seedings establishment?
Lines 95 and 110 are almost similar – may be combined both??
Line 130: leaf surface area/volume – how did you measure?
Line 175: longer total root length = higher total root length.
Line 201: “Drought stress indirectly affected plant height and germination rate by influencing the functional traits of plant leaves and roots” Germination is early stage, and can not be affected by leaf.
Citation 28 is about seeds germination as germination- radicle growth. Please, do not confuse.
Figure 6: I am not sure it is very correct: why root is only adaptive?
Lines 353 – 355: the soil seems to be a very alkaline. What was the reason to use such soil? Did all species tested require this soil type? Medicago is definitely fit, but what about others?
Line 365: moisture gradient- what do you mean as gradient? Inside the pot from surface to bottom??
Line 368: seeds had germinated = seedlings established. Seeds germination is a radicle growth, what is irreversible since seeds resources became to use. Next step can be named as seedlings establishment.
Line 389: “leaf surface area, leaf volume, root length, root surface area, and root volume” – all require precise 3D scan and voxel by voxel analysis. How did you do this??
Line 427 -428: gramineous have a different hormonal source as legumes, so, it is very logic.
Line 430 -435: Leaf as a main source for the root as auxin and carbon, mainly for legumes and less gramineous. Under drought leaf “produce” less resources for root.
Line 437 – 440 – root is an important cytokinin source, required for plant adaptation.
Line 443: among the leguminous plants? Only two plants studied.
Author Response
Comments 1: The current paper studied differential adaptation strategies to drought stress in multiple species based on functional traits. Authors include in analysis Medicago, Agropyron, Elimus and Astragalus and studied it functional traits under different drought conditions in order to use it for soil desertification areas.
Response 1: Thank you for your accurate and concise summary of our research focus and objectives.
Comments 2: Line 47: development of human society- may be too strong conclusion…
Response 2: Thank you for your insightful comment. We understand your concern regarding the use of "the sustainable development of human society and the economy," as it may be too strong a conclusion. To address this, we have revised the sentence to soften the language and better reflect the impact of desertification. The updated sentence now reads: " It remains one of the most severe global ecological problems, jeopardizing both ecological integrity and socio-economic stability in vulnerable areas." We hope this revision addresses your concern and improves the tone of the manuscript. (Page 2, Line 47-48)
Comments 3: Line 68: “also” - to what also??
Response 3: Thank you for pointing out the ambiguous use of "also". We agree with this comment and have therefore deleted the word to improve the logical clarity and flow between the described physiological responses.
Comments 4: Line 69: “inhibited chlorophyll synthesis” per leaf? Per FW?
Response 4: thank you for this valuable comment. We have re-examined references [19-20] and confirmed that the chlorophyll content was measured and expressed as milligrams per gram of leaf fresh weight. This change has been made on Page 2, Line 77 of the revised manuscript. We believe this addresses your concern, and we thank you again for the insightful remark.
[19] Bakhtiari, E.S.; Mousavi, A.; Yadegari, M.; Haghighati, B.; Martínez-García, P.J. Physiological and Biochemical Responses of Almond (Prunus dulcis) Cultivars to Drought Stress in Semi-Arid Conditions in Iran. Plants 2025, 14, 734.
[20] Ma, X.Y.; Zhou, G.S.; Li, G.; Wang, Q.L. Quantitative Evaluation of The Trade-Off Growth Strategies of Maize Leaves under Different Drought Severities. Water 2021, 13, 1852.
Comments 5: Line 79 “traits respond to drought” – traits itself cannot respond.
Response 5: Thank you for your insightful comment. We agree with your observation that "traits" cannot respond by themselves. Accordingly, we have replaced the word "respond" with "change" to accurately reflect this point. The revised sentence on Page 3, Line 98 now reads: "Most studies have emphasized the response of the aboveground parts of plants, while lacking mechanistic explanations of the synergistic changes in both aboveground and underground traits under systematic moisture gradients." Another modification has been made on Page 3, Line 105, where the revised sentence now reads: "How did different plant functional traits change to drought stress? "
Comments 6: Line 86: resistance mechanism- which one- physiological? Epigenetic?
Response 6: Thank you for your valuable comment. In response to your question, the resistance mechanism we refer to in the manuscript is primarily drought resistance mechanism of phenotypic traits, as our study focuses on how plants’ functional traits, growth, and root and leaf characteristics change in response to drought stress. We have clarified this in the manuscript, and the revised sentence now reads: " (2) What were the morpho-structural strategies and the associated drought resistance mechanism of indigenous herbaceous plants under drought stress?" This change has been made on Page 3, Line 105-106 of the revised manuscript. We believe this revision addresses your concern. Thank you again for your insightful suggestion.
Comments 7: Line 96: Seedling germination?? Seedling cannot germinate- seeds can as chromatin activation and primary gradients establishment.
Response 7: We appreciate your valuable feedback. As you pointed out, "seedling germination" is not the correct term. We have revised the sentence to clarify that the focus is on " seedling emergence rate " rather than seedling germination rate. The revised title now reads: "Effects of Drought Stress on Seedling Emergence Rate and Plant Height in Legumes." This change has been made on Page 3, Line 95 of the revised manuscript. Additionally, we have corrected similar errors in other parts of the manuscript, specifically on Page 1, Lines 29-30, Page 3, Lines 112,115,116, 124 and 129, Page 4, Line 131,134, Page 7, Line 224, 229, Page 9, Line 264, 268, Page 13, Lines 418. We believe these revisions address your concern. Thank you again for your helpful suggestion.
Comments 8: Line 104: Changes in germination rate and plant height- at which time point plant height? 3 days? 10 days? What is germination rate – seedlings establishment?
Response 8: We appreciate your valuable comment. In our study, the germination rate refers to the seedling emergence rate, which was determined by recording the number of emerged seedlings in each pot every evening for seven consecutive days after sowing, and calculating the percentage of emerged seedlings on the 7th day. This definition focuses on the early emergence of seedlings, rather than their establishment. The plant height was measured 60 days after sowing, by measuring the distance from the soil surface to the apical bud of three randomly selected standard plants per pot using a measuring tape (accuracy: 1 mm). These details have already been described in Section 4.3 Sample Collection and Analysis (Page 13, Line 421-427). Thank you again for your insightful suggestion.
Comments 9: Lines 95 and 110 are almost similar – may be combined both??
Response 9: Thank you for this thoughtful suggestion. We considered the option of combining these sections to streamline the narrative. Upon careful reflection, we found that maintaining distinct sections for leguminous and gramineous species allows for a clearer presentation and contrast of their specific responses to drought. This structure allows us to more effectively highlight key differences in drought response between these two functionally important plant groups. We appreciate your comment, which prompted us to optimize the manuscript's organization.
Comments 10: Line 130: leaf surface area/volume – how did you measure?
Response 10: Thank you for your question regarding our measurement methodology. In our study, leaf surface area and leaf volume were measured using the Epson Expression 11000XL scanner in combination with WinRHIZO software. The scanner captures high-resolution, two-dimensional images of the leaves, and WinRHIZO software then analyzes these images using image processing algorithms to calculate the surface area. For leaf volume, the software estimates it based on the leaf's projected shape, leaf thickness, and known calibration data, using built-in models to account for the three-dimensional characteristics of the leaf.To further validate our methodological approach, we have added two supporting references [67-68] in Section 4.3 Sample Collection and Analysis, which describe the use of comparable systems and methodologies for leaf trait analysis. We hope this clarifies the measurement process. Thank you again for your valuable feedback. (Page 13, Line 432-434 and Page 21, Line 676-679)
[67] Ding, J.X.; Wang, Q.T.; Ge, W.J.; Liu, Q.; Kong, D.L.; Yin, H.J. Coordination of leaf and root economic space in alpine coniferous forests on the Tibetan Plateau. Plant Soil 2023, 496, 555-568.
[68] Yan, G.Y.; Luo, X.; Liang, C.; Han, S.J.; Han, S.J.; Liu, G.C.; et al. Nitrogen deposition enhances soil organic carbon sequestration through plant-soil-microbe synergies. J Ecol 2025, 113, 2889-2904.
Comments 11: Line 175: longer total root length = higher total root length.
Response 11: Thank you for this precise linguistic correction. As you suggested, we have revised the text by replacing "longer" with "higher" to accurately describe the quantitative difference in total root length. We appreciate your attention to terminological accuracy, which has helped improve the clarity of our manuscript. (Page 6, Line 195)
Comments 12: Line 201: “Drought stress indirectly affected plant height and germination rate by influencing the functional traits of plant leaves and roots” Germination is early stage, and cannot be affected by leaf.
Response 12: Thank you for your valuable comment. We agree that the seedling emergence rate occurs at an early growth stage and cannot be affected by leaf traits or mature root traits. Therefore, after removing the paths from both leaf and root traits to seedling emergence rate, we have re-fitted the SEM in Section 2.3 and made comprehensive revisions to both the results section (Section 2.3) and the discussion section (Section 3.1) to better reflect this relationship. We believe these revisions have improved the accuracy and clarity of our manuscript. Thank you again for your helpful feedback. (Page 7, Line 212-223, Line 231-237, Page 9, Line 279-284)
Comments 13: Citation 28 is about seeds germination as germination- radicle growth. Please, do not confuse.
Response 13: Thank you for pointing out this inaccuracy. We apologize for the confusion and appreciate your clarification. After reviewing Citation 28, we acknowledge that it specifically addresses seed germination in relation to radicle growth. We have reviewed the context of citation and replaced it with more appropriate references [29] that are directly relevant to the specific process we discussed. The manuscript has been updated accordingly. (Page 9, Line 271)
[29] Lu, Y.; Liu, H.; Chen, Y.; Zhang, L.; Kudusi, K.; Song, J. Effects of drought and salt stress on seed germination of ephemeral plants in desert of northwest China. Front. Ecol. Evol. 2022, 10, 1026095.
Comments 14: Figure 6: I am not sure it is very correct: why root is only adaptive?
Response 14: Thank you for this insightful comment regarding Figure 6. We agree that the term "adaptive" may not fully capture the complexity of root responses to drought stress, which can include both active adjustments and passive changes. In response to your feedback, we have revised the title to "Figure 6. Different response strategies of plant leaves and roots under drought stress" to better reflect the nature of these plant organ strategies. We believe this revision more accurately describes the observed changes while maintaining the clarity of our visual summary, and we appreciate your valuable suggestion which has helped improve the precision of our presentation. (Page 12, Line 372)
Comments 15: Lines 353 – 355: the soil seems to be a very alkaline. What was the reason to use such soil? Did all species tested require this soil type? Medicago is definitely fit, but what about others?
Response 15: We thank you for this pertinent question regarding our soil selection. The use of alkaline soil in our pot experiment was a deliberate choice to closely mimic the natural edaphic conditions of the target environment. The six plant species selected for this experiment (Medicago sativa, Astragalus laxmannii, Elymus dahuricus, Agropyron desertorum, Agropyron mongolicum, Agropyron cristatum) were chosen based on field surveys and plant lists, all of which are dominant indigenous herbaceous species in the desertified regions of northwestern China.The soils in these arid and semi-arid regions are typically characterized by alkalinity. Therefore, using an alkaline soil substrate is ecologically relevant and allows us to assess the true drought resistance of these species under conditions they would actually encounter in the field. All selected species are naturally adapted to, or have been widely demonstrated to thrive in, such alkaline soils in their native habitats. Thus, the soil type was appropriate for the entire species set, not only for Medicago sativa. This approach ensures that our findings on drought resistance are directly applicable to species selection for restoration projects in these specific habitats. Thank you again for your helpful feedback.
Comments 16: Line 365: moisture gradient- what do you mean as gradient? Inside the pot from surface to bottom??
Response 16: Thank you for this thoughtful comment. We apologize for the ambiguity in our wording. In this study, “moisture gradient” refers not to the vertical variation of soil moisture within a single pot, but to the six experimentally controlled soil water content levels (15%, 12%, 10%, 8%, 6%, and 4%) that were used to simulate different degrees of drought stress. Once these soil moisture levels were stabilized, seeds were sown in each treatment. We have revised the wording in the manuscript to clarify this point. We appreciate your feedback, which has helped us improve the clarity of our methodological description. (Page 12, Line 400-401)
Comments 17: Line 368: seeds had germinated = seedlings established. Seeds germination is a radicle growth, what is irreversible since seeds resources became to use. Next step can be named as seedlings establishment.
Response 17: Thank you for this precise and instructive comment regarding the distinction between germination and seedling establishment. You are absolutely correct. We have revised the sentence to accurately reflect that the plastic wrap was removed at the stage when seedlings were visibly established above the sand layer. (Page 13, Line 404)
Comments 18: Line 389: “leaf surface area, leaf volume, root length, root surface area, and root volume” – all require precise 3D scan and voxel by voxel analysis. How did you do this??
Response 18: Thank you for this opportunity to clarify our methodology. As noted in our previous response (Response 10), leaf surface area and leaf volume, as well as root length, root surface area and root volume, were measured from high-resolution flatbed scans (Epson Expression 11000XL) and analyzed using WinRHIZO image-analysis software. WinRHIZO analyzes scanned two-dimensional images by skeletonizing root/leaf pixels to obtain total length, and by measuring the projected area and diameter-class distribution to compute surface area and an estimated volume. In other words, root and leaf “volume” reported by WinRHIZO are image-derived estimates based on projected area, length and diameter distributions (using the software’s built-in models). This methodology is consistent with those used in previous studies [67-68], which also applied the same approach for measuring leaf and root traits, further confirming the reliability and comparability of our data. We appreciate your feedback, which has been very helpful in ensuring this methodological point is clearly communicated. (Page 13, Line 432-434 and Page 21, Line 676-679)
[67] Ding, J.X.; Wang, Q.T.; Ge, W.J.; Liu, Q.; Kong, D.L.; Yin, H.J. Coordination of leaf and root economic space in alpine coniferous forests on the Tibetan Plateau. Plant Soil 2023, 496, 555-568.
[68] Yan, G.Y.; Luo, X.; Liang, C.; Han, S.J.; Han, S.J.; Liu, G.C.; et al. Nitrogen deposition enhances soil organic carbon sequestration through plant-soil-microbe synergies. J Ecol 2025, 113, 2889-2904.
Comments 19: Line 427 -428: gramineous have a different hormonal source as legumes, so, it is very logic.
Response 19: Thank you for this positive feedback and for providing the insightful physiological explanation regarding the differing hormonal sources between gramineous and leguminous plants. We agree that this mechanistic perspective strengthens the interpretation of our results. Accordingly, we have incorporated this point into the Discussion section to better explain why legumes exhibited greater susceptibility to drought stress, and have added a supporting reference [34] to this discussion. We sincerely appreciate your constructive input, which has helped us improve the biological interpretation of our findings. (Page 11, Line 358-361)
[34] Waadt, R.; Seller, C.A.; Hsu, P.-K.; Takahashi, Y.; Munemasa, S.; Schroeder, J.I. Plant hormone regulation of abiotic stress responses. Nat. Rev. Mol. Cell Biol. 2022, 23, 680–694.
Comments 20: Line 430 -435: Leaf as a main source for the root as auxin and carbon, mainly for legumes and less gramineous. Under drought leaf “produce” less resources for root.
Response 20: Thank you for this exceptionally insightful comment that significantly deepens our physiological interpretation. We fully agree that our original description lacked this crucial dimension of resource dynamics between organs. You are absolutely correct that the leaf, as a primary source of carbon and auxin, plays an active role in determining root response by modulating resource allocation under drought stress. We have revised the relevant paragraph in the Discussion section to incorporate your excellent point and have added a supporting reference [55] to this discussion. The revised text now explicitly describes the leaf's role not only in conserving resources but also in reducing the allocation of photosynthates and hormonal signals to the roots under drought conditions, which in turn influences and constrains the root's tolerance strategy. This integration provides a more mechanistic and accurate explanation of the coordinated plant response. We are grateful for your suggestion, which has greatly strengthened the narrative of our manuscript. (Page 11, Line 364-369)
[55] Gupta, A.; Rico-Medina, A.; Caño-Delgado, A.I. The Physiology of Plant Responses to Drought. Science 2020, 368, 266–269.
Comments 21: Line 437 – 440 – root is an important cytokinin source, required for plant adaptation.
Response 21: Thank you for your valuable comment. We agree that roots are an important source of cytokinins, which are essential for regulating shoot growth and maintaining plant adaptation under drought stress. We have revised the manuscript accordingly to include this point in the discussion, and have cited a supporting reference [34] to this discussion. We believe this revision provides a more comprehensive description of root functions in drought resistance. Thank you for suggesting this improvement. (Page 11, Line 358-361)
[34] Waadt, R.; Seller, C.A.; Hsu, P.-K.; Takahashi, Y.; Munemasa, S.; Schroeder, J.I. Plant hormone regulation of abiotic stress responses. Nat. Rev. Mol. Cell Biol. 2022, 23, 680–694.
Comments 22: Line 443: among the leguminous plants? Only two plants studied.
Response 22: We thank you for this precise comment regarding the use of "among" when comparing only two leguminous species. We agree that this terminology was inappropriate. We have replaced “among” with “within” to ensure the manuscript reads more formally and precisely. We appreciate your attention to terminological precision, which has helped improve the clarity of our manuscript.

Reviewer 3 Report
Comments and Suggestions for Authors
I have reviewed the above manuscript, which reports a controlled pot experiment comparing six indigenous herbaceous species across six soil moisture levels, measuring growth, leaf and root functional traits, and applying PCA, SEM, and a fuzzy-membership/PCA composite (D-value) to rank drought resistance. The study addresses an important applied topic—selection of species for vegetation restoration in arid regions—and contains a large set of relevant morphological and physiological measurements.
Methodological clarity: Key experimental details (soil characterization, moisture maintenance protocols, exact sampling/replication and whether samples were pooled) need clarification to ensure reproducibility.
Analytical robustness: SEM details and fit statistics are incomplete; uncertainty quantification for PCA-derived weights and the D-value ranking is missing and required.
Interpretation: Some conclusions (e.g., root strategy descriptions) appear inconsistent with reported absolute trait values; authors should present absolute and relative changes and re-evaluate wording.
Extrapolation to field application: Pot constraints should be acknowledged; stronger caution is needed when recommending species for large-scale restoration without field trials.
If the authors address these points (detailed comments are provided in the reviewer report) and supply revised analyses where required, I would consider the manuscript suitable for publication.
Sincerely
Author Response
Comments 1: I have reviewed the above manuscript, which reports a controlled pot experiment comparing six indigenous herbaceous species across six soil moisture levels, measuring growth, leaf and root functional traits, and applying PCA, SEM, and a fuzzy-membership/PCA composite (D-value) to rank drought resistance. The study addresses an important applied topic—selection of species for vegetation restoration in arid regions—and contains a large set of relevant morphological and physiological measurements.
Response 1: We sincerely thank you for the careful evaluation and positive summary of our work. Your recognition of the study’s design and applied significance encourages us to further refine and strengthen the presentation of our findings.
Comments 2: Methodological clarity: Key experimental details (soil characterization, moisture maintenance protocols, exact sampling/replication and whether samples were pooled) need clarification to ensure reproducibility.
Response 2: We sincerely thank you for raising these crucial points regarding methodological clarity. In direct response to these comments, we have augmented the Materials and Methods section to ensure complete reproducibility of our experimental work.
Soil characterization:
We have enhanced the description of soil characterization in the Section "4.1 Test Site and Materials and Methods" to provide better methodological clarity. The revised text now states: " The test soil was collected from the desert-oasis transition zone in Moyu County, Hotan Prefecture of Xinjiang (79°45′48″–79°46′12″ E, 37°29′48″–37°30′12″ N). The soil type is sandy soil and exhibits typical characteristics of a poor substrate in arid regions, including low water-holding capacity, alkalinity, and low organic matter content. The detailed basic properties of the soil are listed in Table 3." This revision provides essential soil characteristics directly in the text while maintaining the detailed data in the table, ensuring better readability and reproducibility of our experimental setup. (Page 12, Line 378-382)
moisture maintenance protocols:
We have enhanced the description of moisture maintenance protocols in the Section "4.2 Experimental Design " to improve methodological clarity. The revised text now states: "Soil moisture was controlled using the weighing method. Pots were weighed every other day using an electronic balance (precision: 0.1 g). The amount of water loss was determined as the difference between the current pot weight and its target weight. Accordingly, water was added to return the pot to its target weight. This protocol maintained the soil water content within a strict tolerance of ±0.5% of the target value throughout the experiment, ensuring the stability and precision of the moisture gradient [66]." This revision clarifies the process used to maintain soil moisture levels, providing essential details in the text while ensuring that the methodology is clear and reproducible. (Page 12, Line 406-412)
Sampling and replication:
We have enhanced the description of sampling methodology and experimental design in the Section "4.3 Sample Collection and Analysis " to provide better methodological clarity. The revised texts now state: " Six soil water gradient treatments were applied to each of the six plant species. Each treatment-species combination was replicated three times, yielding a total of 108 experimental pots." ,"For each replicate pot, three individual plants were randomly selected to measure growth and morphological traits. All measurements were conducted at the individual plant level. "This revision explicitly clarifies the replication structure and sampling protocol, confirming that all statistical analyses are based on independent biological without pooling samples, ensuring the reliability of our results. (Page 13, Line 404-406 ,424-427)
In addition, we have added an explanation in the manuscript regarding the basis for establishing the drought gradient. The drought conditions were Determined based on the Chinese Agricultural Soil Drought Classification Standard (GB/T 32136-2015) and the measured field capacity. The relative water content thresholds for severe, moderate, and mild drought were converted into actual soil water content ranges, and the drought gradients for mild, moderate, and severe drought were accordingly determined. (Page 12, Line 390-394 and Page 14, Line 460-466)
We hope these modifications enhance the methodological transparency of the study and ensure that all key experimental steps can be accurately reproduced.
Comments 3: Analytical robustness: SEM details and fit statistics are incomplete; uncertainty quantification for PCA-derived weights and the D-value ranking is missing and required.
Response 3: We sincerely thank the reviewer for this valuable comment regarding analytical robustness. In response, we have revised the Results and Methods sections to provide more detailed information on the structural equation modeling (SEM), as well as additional clarification on the uncertainty analysis of the PCA-derived weights and the D-value ranking.
Structural Equation Modeling (SEM):
In the revised analysis, the seedling emergence rate was excluded from the final structural equation model. This decision was made because its temporal occurrence during early growth, which temporally preceding and not causally influenced by the later-measured leaf and mature root traits included in the model. Consequently, we updated the model pathways and refitted the SEM. The model's goodness-of-fit is now comprehensively reported using standard indices: χ², df, P (χ² test), GFI, CFI, SRMR, RMSEA. Based on this, we have revised the caption of Figure 5(a). The revised content is as follows: “χ² (assessing the discrepancy between the sample and fitted covariance matrices), df (reflecting the model complexity), χ²/df indicates the overall goodness of fit of the model. A χ²/df value < 3, together with a nonsignificant χ² test (p > 0.05), indicates that the model fits the data well. GFI (Goodness of Fit Index) and CFI (Comparative Fit Index) > 0.90, indicate that the model fits well. SRMR (Standardized Root Mean Square Residual) < 0.08, indicates that the model fits well. RMSEA (Root Mean Square Error of Approximation) < 0.05 indicates that the model fits well.” (Page7, Lines 231–237)
Quantification of the uncertainty in PCA-derived weights and D-value rankings:
To address the uncertainty quantification for both PCA-derived weights and D-value ranking, we employed a comprehensive non-parametric Bootstrap resampling procedure. The analysis was conducted separately for each drought stress level (mild, moderate, and severe). Using the individual pot as the independent sampling unit (n=108 pots), we performed 1,000 random resamplings with replacement. For each of the 1,000 Bootstrap samples, the entire analytical procedure—from PCA through D-value calculation—was repeated, generating distributions of PCA weights and D-value rankings. Based on the Bootstrap distribution, we calculated the standard error and 95% confidence interval for every PCA loading and each plant's D-value ranking. These comprehensive results are now provided in Appendix Table A1-A6. The methodology has been added to the "4.4 Data Processing and Analysis" section as follows: "The uncertainty of both the weights derived from Principal Component Analysis and the subsequent D-value rankings was quantified using a Bootstrap resampling approach with 1,000 iterations. The standard errors and 95% confidence intervals for PCA loadings and D-value rankings are reported in Appendix Table A1-A6." (Page 14, Line 456-460)
The Bootstrap analysis robustly confirms the stability and reliability of our original PCA results and D-value rankings. For the PCA weights, the loadings identified as major contributors to the first principal component (PC1) consistently exhibit high stability with narrow 95% confidence intervals that do not include zero—as exemplified by root volume under mild drought (95% CI: [0.395, 0.446]) and root biomass under severe drought (95% CI: [0.372, 0.434]). Similarly, the D-value rankings demonstrate remarkable stability, particularly for the top-ranked species across all drought stress levels. For instance, Elymus sibiricus maintained its first-place ranking under mild drought conditions with high precision (95% CI: [1, 2]), while Agropyron cristatum consistently ranked first under both moderate and severe drought with equally narrow confidence intervals (95% CI: [1, 3] , equally). This consistent stability in both PCA weights and D-value rankings confirms the robustness of our drought resistance evaluation methodology.
We hope these additional analyses adequately address your concerns regarding uncertainty quantification. Thank you for the constructive feedback that has strengthened our manuscript.
Comments 4: Interpretation: Some conclusions (e.g., root strategy descriptions) appear inconsistent with reported absolute trait values; authors should present absolute and relative changes and re-evaluate wording.
Response 4: We sincerely thank you for this insightful comment. We agree that our initial conclusions regarding root strategies could be more precise in distinguishing between absolute trait values and relative allocation patterns. We have carefully revised the Conclusion section to ensure it accurately reflects our findings.
The revised Conclusion section is as follows: " This study provides insights into the responses of herbaceous plant functional traits to drought stress and their physiological resistance mechanisms. Drought stress was found to directly and indirectly suppress seedling emergence and plant growth. To cope with this stress, plants employ a sophisticated root-leaf coordination strategy. Leaves adopt a resource-conservative strategy, enhancing water-use efficiency and retention capacity through structural regulation. Meanwhile, roots implement a tolerance strategy, improving water acquisition through biomass reallocation and maintaining a relatively higher root–shoot ratio despite the overall biomass reduction. These findings reflect relative rather than absolute trait adjustments, highlighting the plants’ adaptive balance between resource conservation and stress tolerance. Our results confirm that root plasticity serves as a critical factor determining plant drought resistance. Based on a comprehensive evaluation of multiple trait indicators, E. dahuricus, A. cristatum, and M. sativa emerge as highly promising candidates for ecological restoration in arid regions. "
We hope that these revisions improve the alignment between the results and the interpretation, providing a clearer and more balanced description of plant drought-response strategies. (Page 15, Lines 482–499)
Comments 5: Extrapolation to field application: Pot constraints should be acknowledged; stronger caution is needed when recommending species for large-scale restoration without field trials.
Response 5: We sincerely thank you for raising this important point regarding the extrapolation of our pot-based findings to field applications. We fully acknowledge that the inherent constraints of pot experiments, such as limited soil volume and artificially controlled moisture gradients, may influence root architecture development and overall plant responses. Accordingly, we have added a statement acknowledging these limitations at the end of the Conclusion section. The added text reads as follows: "Based on a comprehensive evaluation of multiple trait indicators, E. dahuricus, A. cristatum, and M. sativa emerge as highly promising candidates for ecological restoration in arid regions. However, it is important to acknowledge that these findings are derived from controlled pot experiments, where factors such as restricted soil volume and artificial moisture gradients may affect plant responses. We therefore recommend that future field trials be conducted to validate the performance and ecological adaptability of these species under natural conditions before proceeding with large-scale restoration efforts. We hope that this clarification adequately addresses the reviewer's concern and provides a more cautious and comprehensive perspective on the extrapolation of our pot-based findings. " (Page 15, Lines 494–499)
Comments 6: If the authors address these points (detailed comments are provided in the reviewer report) and supply revised analyses where required, I would consider the manuscript suitable for publication.
Response 6: We sincerely thank the reviewer for their thorough evaluation and constructive feedback. In response, we have carefully addressed all the points raised and made the necessary revisions to the manuscript. We hope that the changes and clarifications provided in the revised manuscript adequately address the reviewer's concerns and contribute to the overall quality of the paper. We are grateful for the opportunity to improve our work and look forward to the possibility of publication.

Round 2
Reviewer 1 Report
Comments and Suggestions for Authors
Authors made substantial changes in the manuscript, which is now significantly improved. Two minor points must be addressed before publication, though.
1) Title is too long and redundant. My suggestion is: “Functional Traits of Herbaceous Plants with Ecological Restoration Potential under Drought Conditions”
2) Despite authors have added information about the species used, they did not provide citations for this. Please, cite the references in the text, to each sentence.
Author Response
We truly appreciate the time and effort you have dedicated to reviewing our work again. Your insightful feedback has been incredibly helpful in further refining the quality of our manuscript. We have carefully addressed each of your comments in this second revision, and the corresponding changes have been highlighted in gray in the manuscript.
Comments 1: Title is too long and redundant. My suggestion is: “Functional Traits of Herbaceous Plants with Ecological Restoration Potential under Drought Conditions”
Response 1: We sincerely thank the reviewer for this constructive suggestion. We agree that the original title was somewhat lengthy. We have adopted the reviewer's proposed title verbatim, as it is more concise and impactful. The new title accurately reflects the core content of our manuscript. The new title is: “Functional Traits of Herbaceous Plants with Ecological Restoration Potential under Drought Conditions”. This revision makes the title more concise and avoids redundancy. Once again, we appreciate your helpful comments, which have contributed to improving the quality of the manuscript. (Page 1, Line 2-3)
Comments 2: Despite authors have added information about the species used, they did not provide citations for this. Please, cite the references in the text, to each sentence.
Response 2: Thank you for pointing this out. We have now included the appropriate citations for the information regarding the species used in the study. The changes are as follows:
“Medicago sativa is a high-quality leguminous forage crop widely used for soil improvement and pasture production [24]. Astragalus laxmannii, a typical pioneer plant for windbreaks and sand fixation, plays a key role in stabilizing shifting sand dunes and facilitating the early stages of vegetation establishment [25]. Elymus dahuricus, an important grass species in the northwestern desert-steppe region, has an extensive underground root network, making it valuable for soil and water conservation projects [26]. Additionally, Agropyron desertorum, Agropyron mongolicum, and Agropyron cristatum are excellent forage grasses widely distributed in northwestern China; their dense root systems effectively stabilize surface soil [27-29].”
We appreciate your attention to this matter and believe this improves the clarity and credibility of the manuscript. (Page 2, Line 84-92)
[24] Fang, Z.H.; Liu, J.N.; Wu, X.M.; Zhang, Y.; Jia, H.L.; Shi, Y.H. Full‑length transcriptome of in Medicago sativa L. roots in response to drought stress. Front Genet. 2023, 13, 1086356.
[25] Gong, W.L.; Ma, L.; Gao, Q.; Wei, B.; Zhang, J.G.; Liu, X.Q.; Gong, P.; Wang, Z.; Zhao, G.Q. Construction of a high-density genetic linkage map and identification of flowering-related QTL in erect milkvetch (Astragalus adsurgens). Crop J. 2022, 10, 1141–1150.
[26] Huang, J.K.; Dai, J.P.; Scarpa, F.; Wang, Y.Q.; Ji, J.N.; Mao, Z. Random root distribution affects the mechanical properties of the soil-root composite and root reinforcement. Catena. 2025, 255, 108896.
[27] Mellish, A.; Coulman, B.; Ferdinandez, Y. Genetic relationships among selected crested wheatgrass cultivars and species determined on the basis of AFLP markers. Crop Sci. 2002, 42, 1662–1668.
[28] Yan, X.; Wu, X.; Sun, F.; Nie, H.; Du, X.; Li, X.; Fang, Y.; Zhai, Y.; Zhao, Y.; Fan, B.; et al. Cloning and functional study of AmGDSL1 in Agropyron mongolicum. Int. J. Mol. Sci. 2024, 25, 9467–9482.
[29] Robins, G.J.; Jensen, B.K.; Waldron, L.B.; Bushman, B.S. Irrigation amount and ploidy affect the turfgrass potential of crested wheatgrass (Agropyron cristatum). Grassl. Sci. 2020, 66, 48–53.

Reviewer 2 Report
Comments and Suggestions for Authors
Thank you for the professional response.
The text is definitely more clear, but still need some corrections:
Line 2: "The Responses of Functional Traits" ??? Traits can not respond.
Line 35: traits are the key factors influencing = adaptation are the key.factors
Line 74: external morphology ?? maybe phenotype??
My best regards!
Author Response
We truly appreciate the time and effort you have dedicated to reviewing our work again. Your insightful feedback has been incredibly helpful in further refining the quality of our manuscript. We have carefully addressed each of your comments in this second revision, and the corresponding changes have been highlighted in gray in the manuscript.
Comments 1: Line 2: "The Responses of Functional Traits"??? Traits cannot respond.
Response 1: We sincerely thank the reviewer for this exceptionally insightful and correct observation. We agree that the phrasing "responses of traits" is logically imprecise. Therefore, we have revised the title throughout the manuscript to more accurately reflect this relationship. The new title is: “Functional Traits of Herbaceous Plants with Ecological Restoration Potential under Drought Conditions”. We hope that these revisions have contributed to improving the overall quality of the manuscript. (Page 1, Line 2-3)
Comments 2: Line 35: traits are the key factors influencing = adaptation are the key.factors
Response 2: Thank you for your insightful comment. We agree with your suggestion that "adaptation" more accurately reflects the intended meaning. Based on your feedback, we have revised the sentence to improve clarity. The updated sentence now reads: “Root system adaptation is the key factor influencing the drought resistance of plants.” We appreciate your valuable input, which has helped to refine the manuscript. (Page 1, Line 34)
Comments 3: Line 74: external morphology ?? maybe phenotype??
Response 3: Thank you for your insightful suggestion. We agree that "phenotype" is a more accurate term to describe the traits of plants in response to environmental stress. Based on your feedback, we have revised the sentence as follows: “However, plants can adjust their phenotype and biomass allocation strategies to adapt to drought environments.” We appreciate your valuable input, which has helped to improve the clarity and accuracy of the manuscript. (Page 2, Line 73)

Reviewer 3 Report
Comments and Suggestions for Authors
The paper have been nicely improved. I suugest acceptance of the manuscript without delay.
Author Response
Thank you very much for your kind words and positive feedback. We greatly appreciate your support and are glad that the manuscript has met your expectations. We are grateful for your time and effort in reviewing our work, which has greatly enhanced the quality of our paper.
